# Testing the role of online group-based supervision for local humanitarian workers following a crisis: A mixed-methods longitudinal study

Gülşah Kurt[1]*, Fatema Almeamari[2], Hafsa El-Dardery[2], Aya Kardouh[2], Scarlett Wong[3], Michael McGrath[4], Louis Klein[4], Ammar Beetar[5], Salah Lekkeh[5], Ahmed El-Vecih[5], Wael Yasaki[5], Ceren Acarturk[2], Dusan Hadzi-Pavlovic[4], Zachary Steel[4], Simon Rosenbaum[4], Ruth Wells[4]

1 School of Psychology, UNSW, Sydney, Australia, 2 Department of Psychology, Koc University, Istanbul, Türkiye, 3 School of Population Health, UNSW, Sydney, Australia, 4 Discipline of Psychiatry and Mental Health, School of Clinical Medicine, UNSW, Sydney, Australia, 5 Hope Revival Organization, Gaziantep, Türkiye

* g.kurt@unsw.edu.au

## Abstract

The 2023 Türkiye-Syria earthquakes caused widespread destruction, leaving millions without access to basic needs and services. Caring for Carers (C4C), an online group-based supervision program for Syrian mental health workers, started three months prior to the earthquakes. This study examined the role of this program in supporting the mental and occupational health of the participants following this crisis. A mixed-method longitudinal design was employed with Syrian mental health workers in Türkiye and Northwest Syria (N = 55, 45.5% female), including 16 months of survey data on psychological distress (Kessler-6) and compassion satisfaction (ProQOL), semi-structured interviews with supervisors (N = 10), and video recordings of supervision sessions (N = 9). A piecewise mixed-effects model within a Bayesian Hierarchical framework was used to assess changes in outcomes across three periods: the active control period (7 months), *pre-earthquake supervision period* (3 months), and *post-earthquake supervision period* (6 months). The thematic analysis was used to analyze the qualitative data from the interviews and supervision sessions. Quantitative findings showed a significant reduction in psychological distress and an increase in compassion satisfaction during *the post-earthquake supervision period* (b = -0.18, error = 0.06, CrI = -0.29, -0.07, b = 0.26, error = 0.04, CrI = 0.18, 0.35, respectively). These changes were significantly different from the active control period (b = 0.21, error = 0.05, CrI = 0.11, 0.31, b = -0.37, error = 0.04, CrI = -0.45, -0.28, respectively) while no significant differences were observed between the active control and pre-earthquake supervision periods. Thematic analysis identified five features of supervision as a safe space and two functions as a source of emotional and practical

**Data availability statement:** The data used in the current study cannot be made publicly available due to ethical restrictions imposed by the UNSW Human Research Ethics Committee (HREC). Specifically, given the small sample size and the circumscribed community of Syrian mental health workers in Northwest Syria, even the de-identified data may contain potentially identifying and sensitive information. Public sharing is not permitted, as participants did not provide consent for public data sharing, and such sharing was not approved by the ethics committee. However, de-identified data can be made available to the interested parties who submit a research proposal detailing the purpose of their inquiry, along with the evidence of an ethics approval from their institution, submitted to the UNSW Human Research Ethics Committee (humanethics@unsw.edu.au). The statistical analysis codes are openly available to ensure transparency and reproducibility.

**Funding:** This research is funded by ELRHA's Research for Health in Humanitarian Crises (R2HC) Program (Grant Number: RG203720) (RW and SR), which aims to improve health outcomes by strengthening the evidence base for public health interventions in humanitarian crisis. R2HC is funded by the UK foreign, Commonwealth and Development Office (FCDO), Wellcome, and the Department of Health and Social Care (DHSC) through the National Institute for Health Research (NIHR). The funding body had no role in the conceptualization; writing of the report; or the decision to submit the report for publication.

**Competing interests:** The authors have declared that no competing interests exist.

support after the earthquakes. Overall, these findings provide evidence for both the protective and promotive role of supervision among Syrian mental health workers, highlighting the key mechanisms through which it may foster resilience and strength in humanitarian settings.

## Introduction

Combining a wealth of quantitative and qualitative data, the present study aimed to examine the stress-buffering role of an online group-based supervision program for psychological distress and compassion satisfaction (a sense of joy and fulfillment derived from a job [1] of Syrian mental health workers in the aftermath of the recent Türkiye-Syria earthquakes. The earthquakes occurred during the implementation of supportive online supervision for mental health workers in the affected area. This provided us with a unique opportunity to test whether supervision could help practitioners cope with the psychological sequelae of earthquakes and increase their resilience.

The Türkiye-Syria earthquakes that struck in early 2023 are considered the worst natural hazards to occur in the European region over the past 100 years [2]. These major earthquakes resulted in widespread destruction, with more than 50,000 people killed, hundreds of thousands injured, and left millions in dire humanitarian need. More than 15 million people in Southeast Türkiye and 8 million in Northwest Syria were affected by the earthquakes. According to official records, approximately 3 million people in Türkiye were internally displaced and migrated to mostly nearby cities [3]. For Syrian refugees already in Türkiye, the earthquakes meant they experienced double displacement, compounding their existing vulnerabilities. Almost half of the Syrian population in Türkiye lived in the earthquake-affected areas prior to the earthquakes [4]. In the immediate aftermath of the quakes, national authorities relaxed the travel restrictions for Syrians for a short period but prohibited their access to aid and assistance provided by local authorities and non-governmental organizations. While some could leave the area, most remained and lacked access to shelter, clean water, and food [5]. The situation was particularly dire inside Syria as basic services and health infrastructure were already insufficient due to the protracted conflict. Aid and assistance were temporarily blocked due to cross-border operation restrictions. Thus, not only did the earthquakes destroy the lives of numerous people, but the subsequent human rights violations and discrimination compounded the existing disadvantages of displaced people [6].

Given the scale of destruction and humanitarian need, the overall disaster relief was insufficient and uncoordinated, largely due to the restrictions imposed by the local government [7]. Relentless community support, including efforts from local actors such as non-governmental organizations, civil societies, and volunteers, played an instrumental role in all relief activities—from rescue activities to providing aid and assistance. Local humanitarian workers were the first to arrive and the last to leave when a crisis happens [8]. Typically, local humanitarian workers are exposed to more traumatic incidents and stressors than international ones, which elevates the risk of experiencing higher levels of psychological distress [9,10]. As part of the same crisis-affected community, local humanitarian workers often struggle with

similar challenges to the community they support. Experiencing crisis-related potentially traumatic incidents makes them susceptible to more psychological distress in response to humanitarian work-related stressors [11]. A recent study with Syrian mental health practitioners supporting Syrian displaced communities in Türkiye and the Syrian border showed that mental health problems are highly prevalent, with at least one-third experiencing depression, anxiety, burnout, and secondary traumatization [12]. The COVID-19 pandemic further compounded the already existing difficulties and worsened the mental health of healthcare workers in and outside Syria [13]. Despite shared traumatic experiences and the stressful nature of their job, Syrian mental health professionals highlight the fulfilling aspect of their work with Syrian individuals as a source of strength, helping them overcome work-related difficulties and fostering resilience. Notably, supervision is considered another essential supportive resource for mental health professionals that helps both manage emotional distress and continue their work with their clients [14].

Supervision is often a neglected but crucial aspect of quality and sustainable mental health care in humanitarian settings [15]. Supervision as a mode of emotional and practical professional support can help practitioners improve their clinical skills and mitigate the risk of developing adverse mental and occupational health outcomes [16–19]. It can also reduce the turnover intention among mental health practitioners by improving job satisfaction [17]. Thus far, limited evidence on the effectiveness of supervision in humanitarian settings is available. Existing studies conducted with participants trained on the Integrated Model for Supervision (IMS), a recent initiative for implementing supervision in humanitarian settings, highlighted the potential benefits of supportive supervision for work (e.g., knowledge, confidence, team cohesion) and mental health outcomes across several humanitarian settings (e.g., Ukraine, Afghanistan, Nigeria, and Jordan) [20–22]. Extending these studies, the present study aims to investigate the effectiveness of delivering supportive supervision on mental and work-related outcomes among mental health workers in a quasi-experimental design by unpacking the changes relative to a control period.

To address this gap, the Caring for Carers (C4C) project [23] aims to test the effectiveness and acceptability of an online group-based supportive supervision program to promote mental health and work-related outcomes among mental health workers in Bangladesh, Northwest Syria, and Türkiye, and improve quality of mental health care provided to Rohingya and Syrian displaced communities in the respective settings. Access and utilization of psychosocial support services are largely restricted in humanitarian settings due to the scarcity of human resources and necessary infrastructure. Recently, online and group delivery of psychological interventions have become widespread as these delivery methods were found to be acceptable, accessible for larger groups, and a cost-effective option alternative to face-to-face delivery in resource-scarce settings [24]. Thus, to circumvent the potential obstacles regarding access and uptake of the program, we chose an online group-based format to provide supportive supervision to mental health workers in our program. A detailed description of the scope and methodology of the project can be found elsewhere [23]. After a 7-month active control period, the supervision program commenced in October 2022, three months prior to the earthquakes, which presented a unique opportunity to investigate its protective role on the mental health of Syrian practitioners enrolled in the program. The current study endeavors to provide insights into the role of supervision in the aftermath of the earthquakes. We hypothesized that the supervision provided after the earthquakes would be associated with significant improvements in psychological distress and compassion satisfaction compared to the active control period, while not necessarily expecting significant differences between the active control and pre-earthquake supervision periods. Qualitative investigations of the first supervision sessions after the earthquakes and interviews with the supervisors will help us understand the key features and roles of the supervision sessions following the earthquakes.

## Methods

### The caring for carers project

The Caring for Carers (C4C) project is a quasi-experimental, community-based participatory study testing the acceptability and effectiveness of an online supportive supervision program. It aims to test the impact of supervision on the well-being of Syrian and Bangladeshi mental health workers and increase the quality of mental health care provided to Syrian and

Rohingya displaced people in respective settings [23]. The C4C project consists of three stages, an active control period and two terms of supervision. There are two implementation sites: Türkiye and Syria, and Bangladesh. The present study utilized data from the Türkiye/Syria site. The humanitarian crisis in Syria represents one of the most protracted crises of all time. Since 2011, political upheavals and conflict situations in Syria have led to the forced displacement of over 12 million people, of whom almost half are internally displaced in Syria [25]. Most of the internally displaced people live in Northwest Syria with extremely limited access to health facilities and basic services [26]. Further, due to the physical distance and geographical position, the majority of Syrian displaced people sought refuge in Türkiye which then became the world's major refugee-receiving country with more than 3 million Syrians living under temporary protection status [25]. Given conflict-related experiences, uncertainty around residency status, and post-displacement stressors, those in Türkiye experience elevated levels of psychological and adaptation problems [27].

The C4C supervision program was designed based on a supportive and reflective supervision modality to address the varied needs of participants in a group format. We aimed not to prescribe a specific supervision model but to allow flexibility in choosing how to facilitate the sessions. We provided supervisors with a handbook including contextual information and suggestions for structuring the sessions. Each participant was asked to present a case study, and supervisors co-facilitated group discussion on the case presented. Case presentations typically focused on complex presentations of psychological problems related to traumatic experiences, loss and grief, family violence and parenting, and challenges in the therapeutic process, such as navigating cultural and gender norms and managing personal boundaries and self-care. Supervisors usually facilitated group discussions on accurate assessment and case formulation, as well as the use of evidence-based strategies such as cognitive-behavioral therapy techniques and exposure through demonstrations and role-plays. The groups also explored ways to address broader social needs related to prolonged conflict and forced displacement through culturally grounded community-based approaches. Additional skills covered included active listening, fostering non-judgmental practitioner responses, and emphasizing practitioner wellbeing and the recognition of effort and care, especially in complex cases where progress may be difficult to observe. There were nine supervision groups, each including 4–6 participants, where one local (Syrian) and one international supervisor co-facilitated the sessions with the interpreting assistance of a trained bilingual research assistant for 90 minutes over Zoom.

## Study design and participants

The present study utilized active control data from Türkiye and Syria (7 months) and the first supervision period broken down into two substages 1) 3-months after the start of the supervision, but before the earthquakes (*pre-earthquake supervision)*, and 2) 6-months after the supervision and the earthquakes (*post-earthquake supervision*) to longitudinally investigate the role of supportive supervision on the mental health of Syrian mental health workers who were located in the earthquake-affected areas. Data used in the present study collected between April 2022 and September 2023, and data was accessed on 28/10/2023. Following the occurrence of the earthquakes, we stopped the data collection for two months until participants indicated their willingness to continue. Thus, the post-earthquake supervision period began two months after the earthquakes. Participants were recruited via the network of the project partner, Hope Revival Organization (HRO), in Türkiye and Syria. HRO sent out invitations to their network of non-governmental organizations offering services to displaced Syrians inside Syria or in Türkiye. Those organizations provided a list of interested mental health workers who provided specialized or non-specialized psychosocial support either in group or individual format at their organizations. Potential participants were contacted by the research assistants and presented with informed consent. Those who agreed to the terms and conditions of the study were asked to fill out the online monthly survey via KoboToolbox [28] during the study. This enabled us to improve the power to detect differences, by using repeated observations over time [29]. Qualitative data was collected to triangulate findings. Video recordings of nine supervision sessions (one from each supervision group) conducted after the earthquakes and 10 interviews with the supervisors were examined to

unpack the key features and functions of supervision in the aftermath of the earthquakes. All supervisors were invited to participate in the interviews. Except for one group, at least one supervisor from each group participated in the interviews.

## Ethics statement

The ethics approval for the C4C project was granted by the ethics committees of the University of New South Wales, Australia (HC210824), Koc University, Türkiye (2021. 395.IRB3.182), and University of Dhaka, Bangladesh (IR211201). The current study was conducted using the dataset from the Türkiye-Syria site of the C4C. All participants provided the written informed consent to participate in the study.

## Positionality

Our team consists of local and international researchers, some with and others without lived experiences of displacement and earthquakes. There is a broad spectrum of representation regarding being an insider living in a conflict and disaster-affected area. Our various positionalities were undoubtedly reflected in how we engaged with research, analyzed, and interpreted the data. Reflecting on our lived experiences and positions, we, as a team, embrace a strength-based, holistic approach to mental health rather than a deficit approach focusing on psychopathologies and weaknesses [30]. Therefore, in the present study, we aimed to gain in-depth knowledge about the role of supervision from multiple perspectives, considering both positive and negative outcome measures to capture a more holistic mental health sequela of the earthquakes. To achieve this, we chose mixed-method research, allowing us to adopt a pragmatic stance that integrates the siloed epistemologies of qualitative (constructivist) and quantitative (positivist) studies and produces contextually relevant and practical results [31].

## Quantitative measures

**Psychological distress.** Kessler-6 Psychological Distress Scale [32] was used to assess the level of general psychological distress among the participants across all stages of the project. It is a 6-item scale rated on a 5-point Likert Scale (1 = all of the time, 5 = none of the time). It has been validated for Arabic-speaking populations [33] and used among conflict-affected Syrian communities previously [34]. Sum scores are calculated with higher scores indicating higher psychological distress. A cut-off of 13 or above is used to indicate clinically significant psychological distress [35]. Cronbach's alpha for this study was 0.926.

**Compassion satisfaction.** The Professional Quality of Life Scale (ProQOL) [1] was developed to measure compassion satisfaction, secondary traumatization, and compassion fatigue among health workers. However, more recent analyses of the factor structure of the ProQOL have called into question the reliability of the secondary trauma and compassion fatigue subscales and supported the reliability of the compassion satisfaction subscale, although debate remains about which items to include [36,37]. Additionally, we measure compassion satisfaction as Syrian members of our research team highlighted the construct as relevant to the lived experience of displacement. Compassion satisfaction was measured using the subscale of the Professional Quality of Life (e.g., satisfied and pleased with the work) rated on a 5-point Likert scale ranging from 1 (very often) to 5 (never). The Arabic version of the scale was previously used [12]. Higher scores indicate higher compassion satisfaction. The internal consistency in the present study was 0.911.

**Potentially traumatic events (PTEs).** The revised version of the Harvard Trauma Questionnaire-Part I [38] was used to identify the potentially traumatic events that the participants have experienced due to the conflict and war situation in Syria. 30 items (e.g., imprisonment, serious injury, and violence) were presented to the participants at the baseline to respond as yes (1) or no (0). High total scores indicate a higher number of PTEs experienced by the participants. The internal consistency was 0.831 in this study.

**Demographic information.** Information on age, gender, profession, years of education, and years in the mental health field were collected.

## Qualitative measures

**Semi-structured interviews.** Online individual semi-structured interviews were conducted in Arabic or English with 10 supervisors (5 local supervisors and 5 international supervisors) to understand their overall experiences as supervisors in the program and inquire about the impacts of the earthquakes on the supervision program and its aftermath role. Open-ended questions were asked about the impact of the earthquakes on program participants, the supervision process, and the role of supervision in the aftermath of the earthquakes. The interviews lasted about one hour to 90 minutes, transcribed, and translated into English if necessary.

**Video recordings of supervision sessions after the earthquakes.** Upon the consent of the participants and supervisors, each supervision session (90 minutes) in the program is recorded, transcribed, and translated into English to obtain detailed information about the activities and processes happening during the session. The trained bilingual research assistants provided translation support during the sessions. The research assistants (who were bi-lingual English-Arabic speakers with training in mental health) also transcribed the full supervision sessions and then directly translated the Arabic spoken parts into English in the final transcriptions.

We analyzed the first post-earthquake session of each of the nine groups to unpack the key features and functions of the supervision session.

## Statistical analysis

**Quantitative data analysis.** We conducted a piecewise mixed-effects model [39] within a Bayesian Hierarchical framework [40]. The piecewise approach enabled us to examine changes over time in the outcome variable during different periods of the study, namely, active control period, pre-earthquake supervision period, and post-earthquake supervision period. This approach is particularly useful for examining the changes in the intercepts (starting point) and slopes (rate of change over time) among different periods representing change or interruption points [41]. We adopted the approach recommended by [39] which uses a mixed effects framework to model the individual growth parameters of participants over time (through repeated measurements) while also examining the population-level effects. The first step is to use graphical displays of responses over time, in addition to information about the timing of an intervention or event, to determine cut-off points to segment the regression lines. Bayesian models are commonly used to deal with small sample sizes as this approach allows the inclusion of informative priors (prior knowledge about the model parameters), which can increase the reliability of results in small sample sizes [42]. We employed a Bayesian approach with Markov Chain Monte Carlo fitting of mixed-effects models that accounted for repeated measures through piecewise regression lines (fixed effects) and a random effect for each of the 55 individuals. First, we presented the results on whether the scores on psychological distress and compassion satisfaction changed significantly over time in each period. Then, we examined whether the changes in outcomes across different periods were significantly different from each other.

We selected informative priors based on prior literature and the treatment effect we expected to see. The mean and variance for the intercepts were based on previous literature in similar populations [12,43,44]. For the piecewise elements, during the active control period, we did not expect to observe a change. During the pre-earthquake supervision period, we also did not expect to observe a change, as we reasoned it was unlikely that the intervention would produce immediate results, in addition, supervision group dynamics would likely take time to develop. Finally, we did expect to see an improvement during the post-earthquake supervision period.

Based on our examinations of the variables, we assumed a normal distribution and generated four chains with 15000 iterations and a warm-up of 4000. Bayesian estimation does not include a significance test as frequentist approaches do. Instead, we determined whether the estimate for a given parameter did not cross the zero mark for the 95% credible

interval. In addition, we ensured that the Rhat value was close to 1 (indicating good convergence). We also specified two hypothesis tests, whether there would be a marked difference in change over time between the active control period and each of the intervention periods. We tested whether controlling for the effect of gender, age, and traumatic experiences improved the model fit and then proceeded with the best fitting model. The analyses were conducted in R studio 4.4.0 with the brms package 2.21.0 [45]. Results are reported in accordance with the Bayesian Analysis Reporting Guidelines [46].

The average completion rate of surveys per time point was 41.95 (5.11), ranging from 55 to 33. Data were determined to be missing completely at random (MCAR) through the use of Little's MCAR test, Chi-Square = 752.896, DF = 727, Sig. = .246. We, therefore, imputed missing data in the outcome variables (compassion satisfaction and psychological distress). We used an intention-to-treat approach, including all cases in the analysis. Multiple imputation was conducted in Mplus (version 8.10) [47] using a regression model (where variables with missing data are regressed on variables without missing data) and analysis type basic. In Mplus, missing data imputation is conducted using Bayesian analysis which has been shown to provide robust imputation with multilevel longitudinal data [48]. We imputed 10 datasets in Mplus (S1 Text) and then imported them into R for analysis. The analytical codes can be found in S2 Text.

**Qualitative data analysis.** Thematic analysis [49] incorporating the elements from Grounded Theory (e.g., axial coding and constant comparison) [50] was performed to engage with and analyze English transcripts from the supervision sessions and the interviews with the supervisors. The same iterative coding process for both supervision sessions and interviews unfolded as follows: initial coding of the data to identify and reflect common patterns, cross-coding to generate codes reflexively, and using sensitizing questions to facilitate focused coding of the data. After developing the codebook, two researchers (author 1 & author 2) independently coded 30% of the data (three supervision sessions and three supervisor interviews) and met to compare codes and finalize the codebook. Discrepancies between the raters were rectified with several discussions and by consulting other researchers in the team. Based on the codes, we identified salient themes and elucidated the relationships between those, using constant comparison and looking for exceptions. The relationships between different components of the qualitative model were visually depicted.

## Results

The sample consisted of 55 Syrian mental health workers (25 female, 45.5%), aged between 26 and 50 years (M = 34.35, SD = 5.17). The majority lived in Syria (89.1%) and were internally displaced (60%). Half of the participants (54.5%) had 1–3 years of work. The mean years of general training were 7.38 (5.60) and 3.91 (2.82) specific to mental health. Of 55 participants, 46 (83.6%) were mental health and psychosocial support (MHPSS) workers who were trained on scalable psychosocial interventions, 7 (12.7%) were psychologists, and 2 (3.6%) were case workers. The mean number of PTEs experienced at the baseline was 8.16 (4.89). The most frequently reported PTEs were witnessed the shelling, burning, and/or razing of residential areas (87.3%), "forced to leave hometown and settle in a different part of the country with minimal services (70.9%)", and "confined to home because of chaos or violence outside" (61.8%). Among the 55 participants, 47 (85.45%) started the first supervision term in the program. Those who dropped out (N = 8) indicated several reasons for discontinuing, such as contract termination, changes in employment, declining to participate, or being unresponsive. There were no statistically significant differences between those who dropped out and those who continued in the program in terms of key demographic characteristics (e.g., gender ($\chi^2(1) = 0.24$, $p = 0.63$ and age (t (53) =1.59, $p = 0.12$) or their initial levels of psychological distress ($t(53) = 0.50$, $p = 0.62$) and compassion satisfaction ($t(53) = 0.39$, $p = 0.70$. 46 out of 47 (97.87%) continued the program after the earthquakes. One participant, unfortunately, passed away due to the earthquakes.

### Piecewise mixed-effects model results

The results of the piecewise mixed models using the imputed data are shown in Table 1 for psychological distress and in Table 2 for compassion satisfaction. The Rhat statistics are close to 1, indicating good convergence, while ESS statistics

are large, indicating the MCMC chains have produced reliable, independent samples. We ran the piecewise models on the observed data first (see supplementary materials for completion rates at each time point and models based on observed data). Model comparison tests showed that adding traumatic events or other demographics did not improve the model, so these were not included in the final model. We then tested the model using imputed data. A sensitivity analysis was conducted to determine whether the specification of weak, moderate or strongly informative priors influenced the results (S3 Text). We saw a similar pattern of results across the different prior specifications, indicating that the priors did not bias the findings. In Table 1 and 2, we present the moderately informative prior results. We note that Bayesian results do not include significance tests, rather we examine the credibility interval and convergence statistics along with posterior predictive checks and model diagnostics.

**Psychological distress.** In Table 1, the coefficient for Time indicates a small (b = 0.11, error = 0.02, CrI = 0.08, 0.15) linear increase in K6 scores for each month during the entire study. The credibility interval does not include zero, indicating a high probability that the effect is positive. After accounting for the effect of Time, the active control period slope does not indicate a probability of additional change over time (CrI = -0.02, 0.08). During the pre-earthquake supervision period, there was no significant change in psychological distress scores (CrI = -0.04, 0.23). Following the earthquake, there was a sudden increase in distress (b = 1.76, error = 0.19, CrI = 1.38, 2.13), with the credibility intervals showing a high likelihood that the effect at this time point increased. We modelled this time point separately as a wild data point. The slope for the post-earthquake supervision period indicates a small reduction in distress (b = -0.18, error = 0.06, CrI = -0.29, -0.07). After accounting for the underlying Time effect, this would be equal to -0.07 (0.11+ (-0.18) per month.

The hypothesis tests comparing the active control period to the two supervision periods showed a probable effect for the active control period compared to the post-earthquake supervision period. When we take into account the underlying increase across time and the active control period coefficient (0.11 + 0.03 = 0.14), this indicates that there was a relative reduction in K6 scores during the post-earthquake supervision period compared to active control (b = 0.21, error = 0.05, CrI = 0.11, 0.31). For the comparison of the active control and the pre-earthquake supervision period, the credibility interval included zero (CrI = -0.18, 0.05), indicating a low likelihood of a difference. Fig 1 shows the predicted K6 score, averaged across the imputed datasets, if the trend during the active control had continued (dotted dark blue line) would have been 7.87:

**Table 1. Changes in psychological distress per study period.**

|  | Estimate | Est. Error | Lower 95% CI | Upper 95% CI | Rhat | Bulk_ESS | Tail_ESS |
|---|---|---|---|---|---|---|---|
| Intercept | 5.35 | 0.43 | 4.50 | 6.20 | 1.00 | 4223 | 8735 |
| Time | 0.11 | 0.02 | 0.08 | 0.15 | 1.00 | 32960 | 61221 |
| Active control period slope | 0.03 | 0.02 | -0.02 | 0.08 | 1.00 | 50927 | 81729 |
| Pre-earthquake supervision period slope | 0.09 | 0.07 | -0.04 | 0.23 | 1.00 | 34783 | 64487 |
| Earthquake occurrence | 1.76 | 0.19 | 1.38 | 2.13 | 1.00 | 41113 | 73602 |
| Post-earthquake supervision period slope | -0.18 | 0.06 | -0.29 | -0.07 | 1.00 | 29749 | 55780 |

**Table 2. Changes in compassion satisfaction per study period.**

|  | Estimate | Est. Error | Lower 95% CI | Upper 95% CI | Rhat | Bulk_ESS | Tail_ESS |
|---|---|---|---|---|---|---|---|
| Intercept | 39.06 | 0.58 | 37.92 | 40.92 | 1.00 | 2737 | 5589 |
| Time | -0.23 | 0.01 | -0.26 | -0.20 | 1.00 | 27853 | 52352 |
| Active control period slope | -0.10 | 0.02 | -0.15 | -0.06 | 1.00 | 38604 | 65935 |
| Pre-earthquake supervision period slope | -0.11 | 0.06 | -0.22 | 0.01 | 1.00 | 29359 | 57701 |
| Post-earthquake supervision period slope | 0.26 | 0.04 | 0.18 | 0.35 | 1.00 | 24263 | 46025 |

PLOS Global Public Health

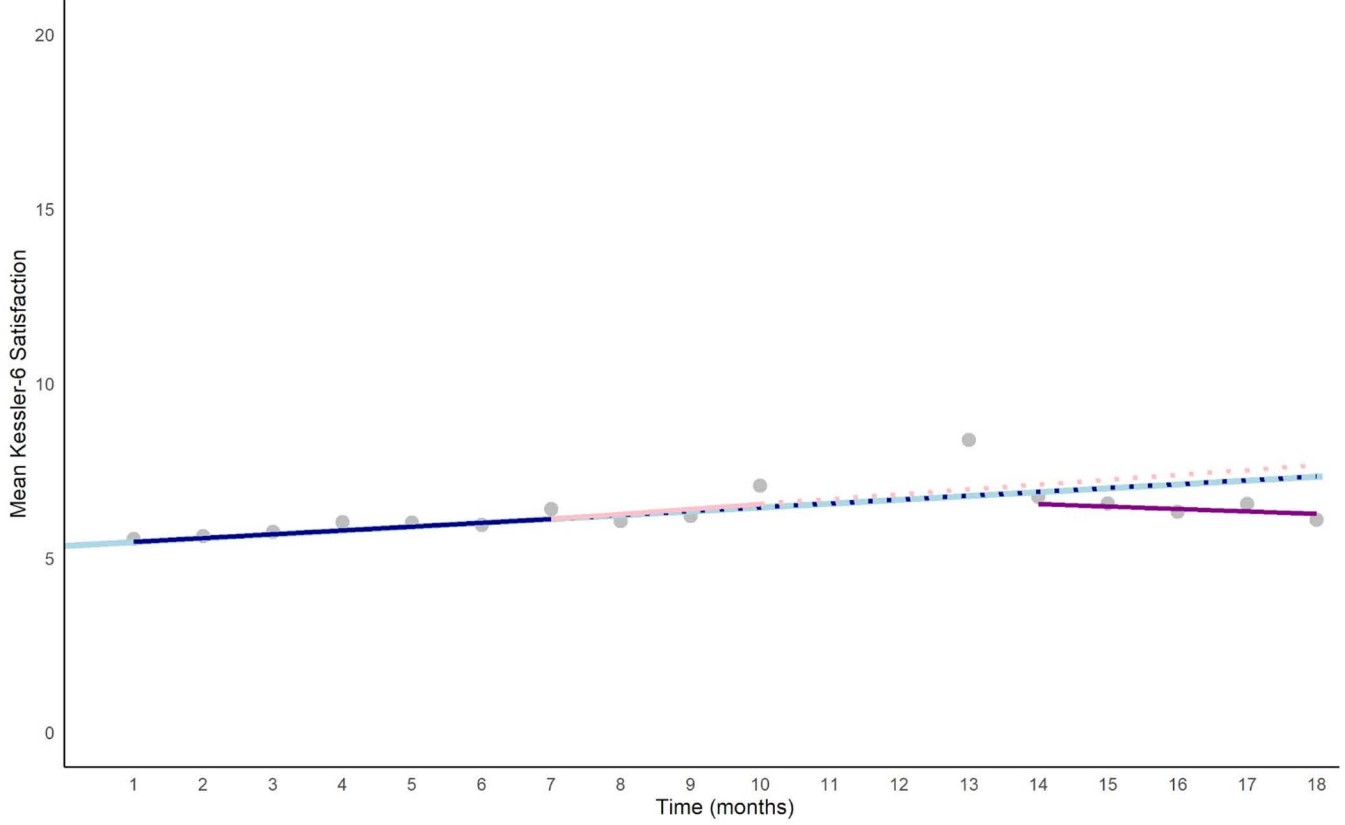

**Fig 1. Psychological distress trajectory over time.**

$Y = \alpha(intercept) + (\beta1(Time) * number\ of\ months + \beta2\ (Active\ Control)) * number\ of\ months$, i.e., $7.87 = 5.35 + 0.11*18 + 0.03*18$.

This is in contrast to the observed mean at time point 18, which was 6.08, despite the underlying increase overtime and the disruption of a major humanitarian crisis.

**Compassion satisfaction.** In Table 2, the Time coefficient indicates a linear small decrease in compassion satisfaction over time (b = -0.23, error = 0.01, CrI = -0.26, -0.20). After accounting for the effect of Time, there was an additional decrease in compassion satisfaction during the active control period (b = -0.10, error = .02, CrI = -0.15, -0.06). When combined with the Time coefficient, the estimated decrease in compassion satisfaction during this period is 0.33 (-0.23 + -0.1). During the pre-earthquake supervision period, there was no additional decrease in compassion satisfaction (CrI = -0.22, 0.01). However, during the post-earthquake supervision period, there was a significant increase in compassion satisfaction (b = 0.26, error = 0.04, CrI = 0.18, 0.35). After taking the effect of Time into account, the increase in compassion satisfaction would be equal to 0.03 per month.

The hypothesis tests comparing the active control period to two supervision periods showed a probable effect for the active control period compared to the post-earthquake supervision period (b = -0.37, error = 0.04, CrI = -0.45, -0.28), but not for the active control period to pre-earthquake supervision period (CrI = -0.1, 0.1).

Fig 2 shows the predicted compassion satisfaction score, averaged across the imputed datasets, if the trend during the active control had continued (dotted dark blue line) would have been:

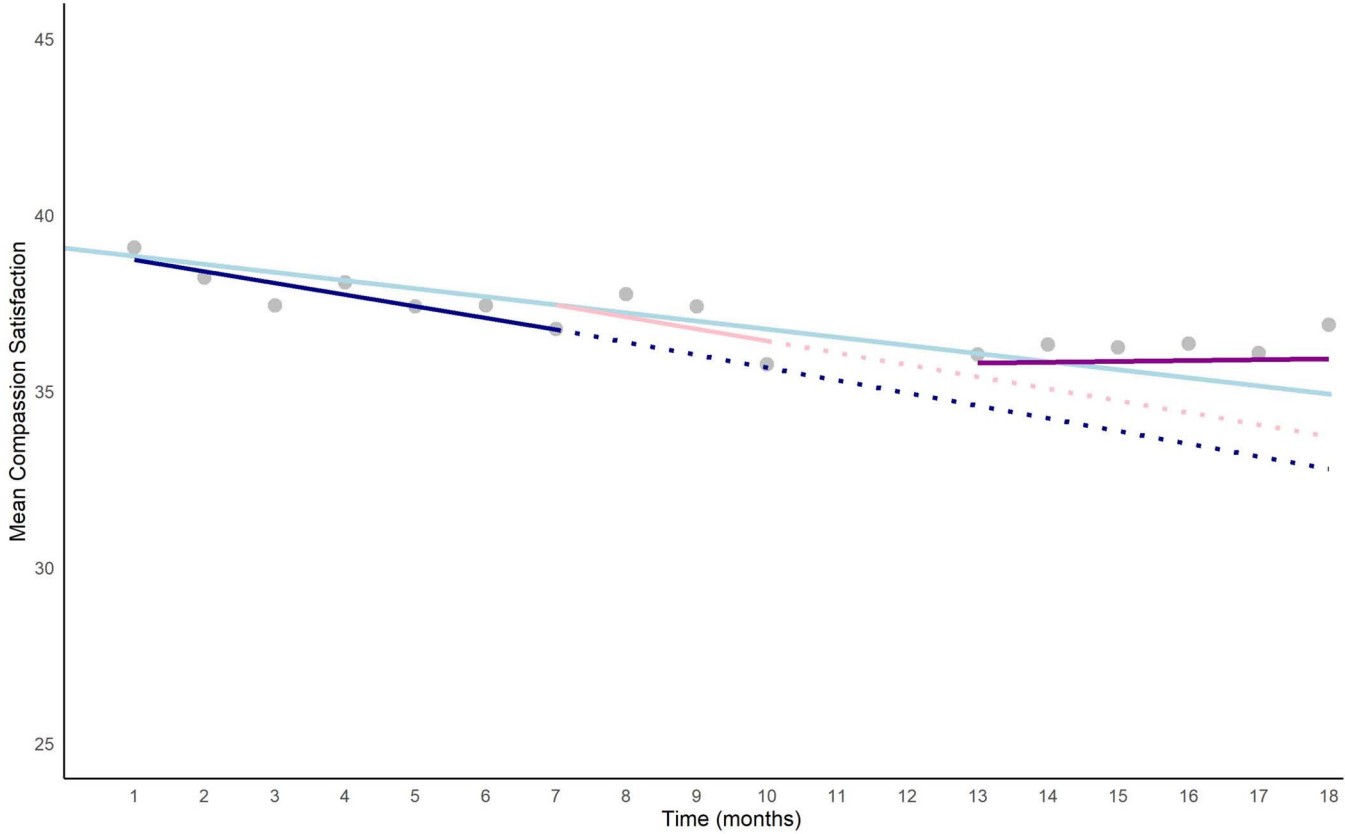

**Fig 2. Compassion satisfaction trajectory over time.**

$Y = α(intercept) + (β1(Time)*number\ of\ month + β2\ (Active\ Control)) * number\ of\ months$, i.e., 33.12 = 39.06 + (-0.23) * 18 + (-0.10)*18.

This is in contrast to the observed mean at time point 18, which was 36.9, despite the underlying increase overtime and the disruption of a major humanitarian crisis. That is, during post-earthquake supervision, the deterioration in compassion satisfaction stopped.

## Qualitative results

We identified several themes from supervision sessions (48% female) and supervisor interviews (50% female) delineating five features of supervision: 1) genuine presence of the supervisors; 2) giving space; 3) empathetic responding; 4) connecting practitioners to each other; and 5) restoring a sense of normality, and two functions of supervision 1) providing emotional support, and 2) providing practical support. Interviews with supervisors provided insights into the key features of supervision following the earthquakes while supervision sessions allowed us to understand how practitioners utilized these sessions to receive emotional and practical support as the examples unfolded. Supervision served as a safe space for practitioners to share their experiences and support each other. We used the "*umbrella*" metaphor to illustrate the stress-buffering role of supervision following the earthquakes as it helped practitioners to feel supported while navigating personal and professional challenges. Fig 3 depicts the relationship between the identified themes.

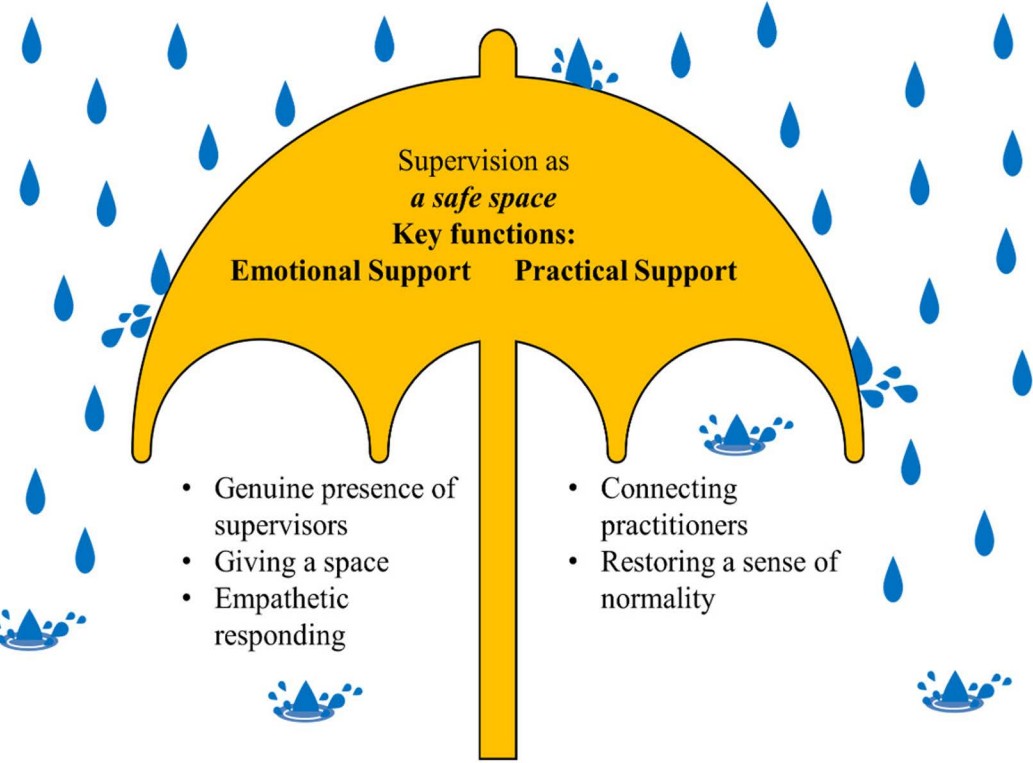

**Fig 3. The relationship between the key features and functions of supervision.**

**Key features of supervision from interviews with supervisors.** From the interviews with the supervisors, it became clear that they were unprepared for such a sudden and destructive emergency. Yet, they wanted to continue their sessions to show their support and care for the practitioners. They highlighted their genuine willingness to be present for the practitioners, even if they did not know how to respond to the crisis.

"*So, for them to know, they just had some people who were looking out for them at that time. Not much we could do, obviously, but just to be available to them and to reflect on that. So, I think we had two or three sessions that were pretty much just 'This is what's going on for you at the moment. How are you coping with that?'"* (Male International Supervisor, Interview).

"*It says something, you know, it says that they really believe that somebody is really caring, somebody is really there for them. For nothing in return.*" (Female Syrian supervisor, Interview).

Although the practitioners were in the field one day after the earthquakes and derived strength from their work with their community, the supervisors were cognizant of the challenges that practitioners might have faced. Consequently, they emphasized the importance of creating a space where the practitioners could share their emotions and feelings following their first-hand experiences of the earthquakes and their work with the affected individuals. While doing so, supervisors empathetically engaged with the practitioners, responding thoughtfully when they shared their personal experiences. Further, it was evident in the supervision sessions that holding this space also helped practitioners to stay connected with and support each other.

"*It was more of them talking a bit about their family situation, their own safety, their difficulty with attending work, and what that meant to them. Some of their family members and extended friends just died, some losses and things there. So that was probably a point of connection.*" (Male International supervisor, Interview).

"*Practitioners want to share, because they can't share it with their families, maybe with their parents, maybe with their friends, because they have this same experience. And they don't want to make it difficult for each other. So, they need someone from outside to hear from the professional ears.*" (Female Syrian supervisor, Interview).

Lastly, supervision was one of the few things that remained constant in the lives of the practitioners. Having supervision sessions after the earthquakes helped provide a sense of normality and stability amidst the volatility and chaos following the earthquakes. Supervisors were quite surprised by the practitioners' desire to return to normality and focus on practical matters, such as how they could assist the affected individuals in their communities.

"*Having the supervision continue gave a sense of normality, that there was something that was hoped for that could go back to normal…Everything changed. And this was a constant thing that stayed the same.*" (Male International Supervisor, Interview).

Altogether, these five key features established supervision as a safe space for practitioners in the aftermath of the earthquakes, where they could receive various forms of support. The primary functions of the supervision were to offer emotional and practical support to the practitioners.

**Key functions of supervision from video-recordings of supervision sessions.** We identified two key functions of supervision based on the supervision sessions which provided examples of how practitioners utilized those sessions to receive emotional and practical support.

**Emotional support:** In the videos of the supervision sessions, practitioners openly shared their experiences and emotions related to the earthquakes such as helplessness, hopelessness, sadness, numbness, and concentration problems. Several practitioners mentioned that the earthquakes compounded the impacts of traumatic experiences due to the conflict and instability in Syria and led to exacerbation of the difficulties experienced by the Syrian community.

"*I'm afraid that after all the traumatic situation I went through ended up with feeling sad or freezing. My feelings don't respond anymore and also I have noticed myself my concentration is very low at this time.*" (Female Syrian Practitioner, Supervision Session).

"*If these earthquakes happened in a stable place, it would have been different because here there is already an economic crisis and it's a war zone and we are already suffering from many things and the earthquakes happened it's been double damage for us.*" (Male Syrian Practitioner, Supervision Session).

Practitioners also discussed several coping strategies that they employed to deal with the psychological sequela of the earthquakes, such as religious and spiritual coping, positive reappraisal, helping others, and engaging in self-care activities (e.g., sports and hobbies).

"*The situation is beyond the tolerance of human mind. You cannot say anything other than what pleases God. We belong to God and to Him shall we return, and we try to stand by our brothers and people who are most affected, even there are people who cannot bear it even if they were not affected. But God has wisdom in everything.*" (Male Syrian practitioner, Supervision Session).

After practitioners shared their personal experiences, supervisors engaged in normalizing the practitioners' experiences and validated their feelings and emotions.

"*This is a natural disaster, and it is devastating. And of course, you're going to be reacting to that, because it is still ongoing right now. So, your symptoms are still going to be there for as long as this is still an issue.*" (Female International Supervisor, Supervision Session).

Most of the practitioners expressed a strong sense of gratitude and appreciation for the emotional support they received from their group and supervisor. They emphasized the positive impacts of support and listening.

"*It was very comfortable, and we received great moral support. I thank you with all my heart.*" (Male Syrian Practitioner, Supervision Session)

"*About the session, it is more than wonderful to listen and support each other.*" (Male Syrian Practitioner, Supervision Session)

**Practical support:**  Practitioners also utilized supervision sessions to receive practical support, where they discussed the needs of at-risk groups following the earthquakes (e.g., women, children, and elderly) and how to support them. For instance, in one supervision group, they extensively discussed the psychological needs of children and the necessity of restoring children's sense of safety and normality.

"*I think the hardest thing for the children is that until the parents and carers feel safe- some sense of safety, the children can't feel safe. There's no way to help the children you know, and so you have to really provide that support to the parents first, but if there are still aftershocks and still terrible things going on, it's really hard to do that.*" (Male International Supervisor, Supervision Session)

Supervisors provided psycho-education around common psychological reactions in the aftermath of the crises and when to employ evidence-based strategies at various intensities to address the needs of affected individuals. For instance, several groups discussed at length the importance of offering psychological first aid and community support during the early stages of the crisis. They also highlighted the fact that not all affected individuals develop trauma-related mental health problems, thereby requiring specialized treatment. The groups also discussed specialized treatment approaches for at-risk groups.

"*Our plan for psychological first aid was just to see what needs there are, to help connect them, so we made like a map of the services around the area. And during work, we saw how psychological first aid is important, especially on the first day. The reason is that most people need someone with them to support them and listen to them and connect them to information that they need. Because during the shock, it's hard to know about these things.*" (Male Syrian Practitioner, Supervision Session)

"*…there are many things that expose people to traumatic events, but it's not necessary for everyone to develop depression, PTSD, and other disorders. So, community support has a major role in limiting disorders and helps in recovery... It supports people a lot and seeing that we are all equal helps us overcome on the long-term, individual disorders that may develop after some period…So the presence of community support and solidarity that we witnessed. Like some people improved or restored their relationships with others, all of this has a far-reaching effect and reduces psychological disorders on the long-term.*" (Female Syrian Supervisor, Supervision Session).

## Discussion

We investigated the role of supportive supervision on mental health and compassion satisfaction of local mental health workers in a complex humanitarian emergency affected by conflict and disaster. Our unique study design allowed us to longitudinally examine the changes in the levels of psychological distress and compassion satisfaction over 16 months and unpack the key features and functions of the supervision during this period. The study findings corroborated our assumptions. The results demonstrated the potential effectiveness of the supervision program in alleviating psychological distress and increasing compassion satisfaction among Syrian mental health workers supporting displaced Syrian communities in Türkiye and Northwest Syria in the aftermath of the earthquakes.

Participants were distressed at the initial onset of the program, with a slight increase in psychological distress and a decrease in compassion satisfaction. Although the pre-earthquake supervision period was not significantly different from the active control period, we found the supportive role of supervision following the earthquakes. These findings suggest a potential incubation period in the first three months of the supervision program to become effective, and its potentially protective and promotive role by not only mitigating the adverse impacts of the earthquakes but also improving mental and occupational health. While prior research documented the role of clinical supervision in reducing psychological distress and improving occupational outcomes, such as increasing job motivation and staff retention, both in low [16] and high resource settings [19], our study extends this literature by demonstrating that supervision may also play a stress-buffering role in the event of a large-scale humanitarian emergency. This is particularly relevant in pernicious and volatile settings where mental health workers are at a higher risk of experiencing multiple traumatic experiences and stressors. Existing studies in humanitarian settings have also emphasized the importance of supervision for both the well-being of humanitarian workers and the quality of service delivered [51]. However, most of this research is either cross-sectional, showing the association between perceived supervisor support and outcomes [52], or focuses on the role of supervision in enhancing competence rather than well-being [53]. More recently, studies in complex humanitarian settings (e.g., Ukraine, Afghanistan, Jordan, and Bangladesh) [22] have examined the acceptability, effectiveness, and scalability of a multi-component supervision training program by assessing changes in knowledge, confidence, and psychological outcomes following training completion, rather than directly testing the impact of receiving ongoing supervision for humanitarian workers. Thus, the present study provides initial empirical evidence on the effectiveness of online, group-based supervision in supporting the mental health of humanitarian workers.

Considering that participants experienced the earthquakes firsthand, the notable increase in psychological distress afterward was not surprising. The qualitative findings from the supervision sessions supported this increase by revealing that participants perceived this catastrophe as both traumatizing and compounding their existing difficulties. Yet, among other coping strategies mentioned by the participants, helping others in their community might explain why there was no sudden decline in compassion satisfaction, which, in contrast, was increasing following the earthquakes. The presence of heightened psychological distress and compassion satisfaction is in line with the existing studies showing that humanitarian workers can experience significant psychological distress while simultaneously finding meaning and fulfillment in helping others from the same community [12,14].

The qualitative inquiry into the supervision sessions and the interviews with supervisors enabled us to explore the key features and functions of the supervision in-depth and complement the quantitative results. The findings highlighted the main role of supervision as being a safe and supportive space for practitioners where they discussed their personal and work-related challenges and received both emotional and practical support from their peers and supervisors. This aligns with extensive research showing the supportive function and practical and emotional benefits of supervision for mental health professionals [54,55]. Yet, it is important to note that there was substantial variance in both psychological distress and compassion satisfaction that was not explained by the supervision itself. Thus, taking both quantitative and qualitative findings together, we likened the stress-buffering role of supervision to an umbrella- though it cannot provide full protection from the rain, it prevents practitioners from getting completely wet on rainy days. We do not expect any program to

completely shield people from the impacts of these kinds of events. Thus, we are interested in what supports their coping resources and resilience despite the hardships that they must contend with. This mixed-method investigation involving multiple sources of data showed us that continuous supervision can be one of the ways to support the mental health of local humanitarian workers.

The intricate loss of social, material, and personal resources following the Türkiye-Syria earthquakes foreshadows the necessity of effective and sustainable psychological recovery planning for the earthquake survivors [7]. It is still hard to gauge the psychological impacts of the destruction and loss while access to basic needs and services remains limited [56]. Multiple humanitarian crises are likely to occur in low-resource settings where disadvantaged groups such as displaced people are disproportionally affected [57]. The earthquakes, therefore, present an additional challenge for the conflict-affected Syrian community inside Syria as public health systems, already destroyed by ongoing conflict, are insufficient to respond to excessive needs [58]. Thus, the sustainability of mental health and psychosocial support activities relies on local mental health workers. To ensure the effective and continued delivery of these activities, it is crucial to strengthen the capacity of local mental health workers and support them through supervision [59]. In this regard, we hope that our findings will serve as a catalyst for decision-makers at the organizational and policy levels to incorporate supervision into routine mental health care in complex humanitarian settings. For humanitarian organizations, this means placing supervision at the centre of their MHPSS portfolio, not as an ad-hoc to trainings, but as an essential component of routine service delivery to ensure the provision of quality care that supports both mental health workers and service users. To enable an effective and sustained MHPSS response, supervision must be integrated into all programs with ongoing funding allocated throughout the implementation of MHPSS programs. These structural changes to funding, policy, and program design require a coordinated effort and strong commitment from multiple stakeholders, such as international agencies, donors, implementing organizations, and mental health workers to establish supervision as a standard practice that safeguards staff well-being and upholds the quality of MHPSS services.

The practical implications of our findings can extend beyond the current crisis setting and can be informative for other complex humanitarian contexts affected by conflict or natural disasters. Despite differences in the origin of the crisis across diverse settings, humanitarian settings often grapple with a lack of necessary health facilities, infrastructures, and qualified human resources. The online and group-based delivery of supervision presents a potentially effective and scalable approach to support the mental health of humanitarian workers and contributes to strengthening the local workforce across diverse settings. Since the scientific investigation of supervision in humanitarian settings is at a nascent stage, the present study is one of the first to systematically test the potential role of supervision in a complex humanitarian setting. Our findings provide a scaffold for future studies to build on and examine different types and modalities of supervision in other contexts.

The present study has some limitations. Despite the breadth and depth of the data included in this study, our sample was relatively small, and the majority of participants were living in Syria. Our sample characteristics were comparable to those of other studies with Syrian humanitarian workers in the same context, with a difference in the living place (e.g., Hamid et al., 2020). Therefore, it is not possible to extrapolate the current findings to Syrian mental health workers who provide mental health care in other regions of Syria and Türkiye. The male predominance in our sample might have also over-represented their experiences, although we believe that we mitigated this risk by examining the qualitative data from both female and male participants in detail. The current findings only provide evidence in favor of the online and group modality of supervision, which necessitates testing the comparative effectiveness of other delivery methods, such as face-to-face supervision. It is also important to note that the quasi-experimental design of the study does not allow us to draw causal conclusions. While our findings suggest a potential stress-buffering role of the supervision program, these results should be interpreted with caution and replicated in future studies using other designs such as randomized controlled trials. Further, it is possible that social desirability influenced the qualitative feedback from the supervisors. To address this issue, we triangulated quantitative and qualitative data from multiple resources to ascertain our conclusion

about the stress-buffering role of supervision. Our position as local and international researchers, with and without experiences of displacement and natural disasters, might have impacted our impartiality in interpreting the results and making inferences about the role of supervision. However, we strongly believe that our diverse positionality provided us with several advantages, such as gaining insights into the importance of adopting a strengths-based approach rather than a deficit approach when examining the psychological sequelae of humanitarian emergencies. Lastly, the long-term impact of supervision on mental health and compassion satisfaction of practitioners following the earthquakes should be subject to scientific inquiry to determine whether the stress-buffering role is maintained over time.

## Conclusion

The present study demonstrated the protective and promotive role of an online supervision program for the mental health of Syrian mental health workers following the 2023 earthquakes in Türkiye and Syria. The current findings indicate that the supervision program enhanced participants' coping capacity to deal with the psychological impacts of the earthquakes. Qualitative findings expanded the results of the quantitative investigation, revealing the key features and functions of supervision during this time. Altogether, these findings highlighted the importance of integrating supervision into routine mental health care to support the mental health of local workers, thereby strengthening the capacity of local mental health care systems.

## Supporting information

**S1 Text. Mplus Code for Multiple Imputation.**
(PDF)

**S2 Text. Example R code for Bayesian hierarchical model estimation.**
(PDF)

**S3 Text. Outputs for Bayesian hierarchical model testing.**
(PDF)

## Author contributions

**Conceptualization:** Gülşah Kurt, Louis Klein, Ammar Beetar, Salah Lekkeh, Ahmed El-Vecih, Wael Yasaki, Ceren Acarturk, Zachary Steel, Simon Rosenbaum, Ruth Wells.

**Data curation:** Louis Klein, Dusan Hadzi-Pavlovic, Ruth Wells.

**Formal analysis:** Gülşah Kurt, Fatema Almeamari, Aya Kardouh, Michael McGrath, Louis Klein, Dusan Hadzi-Pavlovic, Zachary Steel, Ruth Wells.

**Funding acquisition:** Scarlett Wong, Ammar Beetar, Salah Lekkeh, Ceren Acarturk, Dusan Hadzi-Pavlovic, Zachary Steel, Simon Rosenbaum, Ruth Wells.

**Investigation:** Gülşah Kurt, Fatema Almeamari, Hafsa El-Dardery, Scarlett Wong, Michael McGrath, Louis Klein, Ammar Beetar, Salah Lekkeh, Ahmed El-Vecih, Wael Yasaki, Ceren Acarturk, Dusan Hadzi-Pavlovic, Zachary Steel, Simon Rosenbaum, Ruth Wells.

**Methodology:** Gülşah Kurt, Fatema Almeamari, Hafsa El-Dardery, Scarlett Wong, Louis Klein, Ammar Beetar, Salah Lekkeh, Ahmed El-Vecih, Wael Yasaki, Ceren Acarturk, Dusan Hadzi-Pavlovic, Zachary Steel, Simon Rosenbaum, Ruth Wells.

**Project administration:** Gülşah Kurt, Fatema Almeamari, Hafsa El-Dardery, Aya Kardouh, Michael McGrath, Louis Klein, Salah Lekkeh, Ahmed El-Vecih, Wael Yasaki, Ceren Acarturk.

**Resources:** Gülşah Kurt, Fatema Almeamari, Hafsa El-Dardery, Aya Kardouh, Scarlett Wong, Salah Lekkeh, Ahmed El-Vecih, Simon Rosenbaum.

**Software:** Ruth Wells.

**Supervision:** Ruth Wells.

**Writing – original draft:** Gülşah Kurt.

**Writing – review & editing:** Fatema Almeamari, Hafsa El-Dardery, Aya Kardouh, Scarlett Wong, Michael McGrath, Louis Klein, Ammar Beetar, Salah Lekkeh, Ahmed El-Vecih, Wael Yasaki, Ceren Acarturk, Dusan Hadzi-Pavlovic, Zachary Steel, Simon Rosenbaum, Ruth Wells.

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
