## [Decision Letter · Decision Letter 0]

1 Jul 2025

PGPH-D-25-01030

“Everything changed, this was the constant thing that stayed the same.” A mixed-method investigation of the role of online group-based supervision for Syrian mental health workers in the aftermath of the Türkiye-Syria earthquakes.

Dear Dr. Kurt,

Thank you for submitting your manuscript to PLOS Global Public Health. After careful consideration, we feel that it has merit but does not fully meet PLOS Global Public Health’s publication criteria as it currently stands. Therefore, we invite you to submit a revised version of the manuscript that addresses the points raised during the review process.

Please address the minor revisions requested with particular attention to the conciseness of language.

We look forward to receiving your revised manuscript.

Kind regards,

Hani Mowafi, M.D., M.P.H.

Academic Editor

Journal Requirements:

Additional Editor Comments (if provided):

Please address the minor revisions requested including making some language more concise.

Reviewers' comments:

Reviewer's Responses to Questions

**Comments to the Author**

1. Does this manuscript meet PLOS Global Public Health’s publication criteria?

Reviewer #1: Yes

Reviewer #2: Yes

2. Has the statistical analysis been performed appropriately and rigorously?

Reviewer #1: Yes

Reviewer #2: Yes

3. Have the authors made all data underlying the findings in their manuscript fully available (please refer to the Data Availability Statement at the start of the manuscript PDF file)?

Reviewer #1: No

Reviewer #2: Yes

4. Is the manuscript presented in an intelligible fashion and written in standard English?

Reviewer #1: Yes

Reviewer #2: Yes

Reviewer #1: This manuscript presents a timely and important mixed-methods study investigating the stress-buffering role of online group-based supervision for Syrian mental health workers following the 2023 Türkiye-Syria earthquakes. The topic is highly relevant to global public health, particularly in humanitarian and disaster-affected settings. The mixed-methods approach is a strength, offering both quantitative evidence of the supervision program's effectiveness and qualitative insights into its key features. The authors have utilized a quasi-experimental design, taking advantage of a unique natural experiment. The findings suggest that post-earthquake supervision is associated with significant improvements in psychological distress and compassion satisfaction. While the study has significant strengths, there are areas where clarity, detail, and discussion could be enhanced to improve its impact and suitability for publication.

-Title: The title is evocative and effectively captures the essence of the study's qualitative findings. However, the first part ("Everything changed, this was the constant thing that stayed the same.") could be integrated more smoothly with the descriptive part to maintain a professional academic tone. Consider rephrasing for conciseness while retaining its impact. Consider a more concise and direct title while retaining the essence of the mixed-methods approach and key findings.

-Abstract: The abstract provides a good overview of the study. Ensure all quantitative results mentioned (e.g., specific changes in distress/satisfaction) are precisely stated and accompanied by their statistical significance.

-Supervision Program (C4C): More details could be provided on the content of the supervision sessions. While it's mentioned that supervisors were given a handbook and participants presented case studies, elaborating on the typical topics, skills addressed, or therapeutic approaches discussed would strengthen the methodological description.

-Discuss attrition rates and how missing data were handled (e.g., were dropouts related to earthquake impacts?

-The "stress-buffering" effect is compelling, but causality cannot be inferred due to the quasi-experimental design. Frame conclusions more cautiously.

-Contrast the study’s results with prior literature on supervision in non-humanitarian settings, and/or other humanitarian settings to highlight uniqueness.

-Strengthen Implications: Expand on the practical and policy implications of your findings, particularly for humanitarian organizations and mental health support programs.

Overall, this is a well-designed and executed study with important findings. Addressing these points will significantly enhance its clarity, rigor, and impact for publication in an academic journal.

Reviewer #2: This paper meets the criteria for publication in PLOS Global Public Health. The paper presents preliminary evidence for the use of an online supervision initiative to support the mental health workers providing care in aftermath of the Türkiye-Syria earthquakes. This is particularly important given the increase in frequency of natural disasters both in the region and internationally, which will require innovative and novel methods of providing and maintaining acute and chronic mental health care provisions. The use of mixed methods to present the data is appropriate with qualitative analyses being appropriate to further explore and contextualize quantative data. They are also reproducible. Overall, the data and subsequent analyses support the claims that The Caring for Carers (C4C) project may improve psychological distress and compassion satisfaction among Syruan mental health workers in this context, albeit using a small sample size (n=55)to support these claims. The paper is written succinctly, although the abstract of the paper may benefit from simpler, more technical wording.

**Do you want your identity to be public for this peer review?** For information about this choice, including consent withdrawal, please see our Privacy Policy

Reviewer #1: No

Reviewer #2: **Yes: ** Harun Khan

---

## [Editor Report · Decision Letter 1]

30 Jul 2025

Testing the role of online group-based supervision for local humanitarian workers following a crisis: a mixed-methods longitudinal study

PGPH-D-25-01030R1

Dear Dr Kurt,

We are pleased to inform you that your manuscript 'Testing the role of online group-based supervision for local humanitarian workers following a crisis: a mixed-methods longitudinal study' has been provisionally accepted for publication in PLOS Global Public Health.

Best regards,

Hani Mowafi, M.D., M.P.H.

Academic Editor

Thank you for your thoughtful responses to the reviewer comments